# Unified Insights: Harnessing Multi-modal Data for Phenotype Imputation via View Decoupling

**Qiannan Zhang, Weishen Pan, Zilong Bai, Chang Su, Fei Wang**[*]
Weill Cornell Medicine, Cornell University
{qiz4005,wep4001,zib4001,chs4001,few2001}@med.cornell.edu

## Abstract

Phenotype imputation plays a crucial role in improving comprehensive and accurate medical evaluation, which in turn can optimize patient treatment and bolster the reliability of clinical research. Despite the adoption of various techniques, multi-modal biological data, which can provide crucial insights into a patient's overall health, is often overlooked. With multi-modal biological data, patient characterization can be enriched from two distinct views: the biological view and the phenotype view. However, the heterogeneity and imprecise nature of the multi-modal data still pose challenges in developing an effective method to model from two views. In this paper, we propose a novel framework to incorporate multi-modal biological data via view decoupling. Specifically, we segregate the modeling of biological data from phenotype data in a graph-based learning framework. From the biological view, the latent factors in biological data are discovered to model patient correlation. From the phenotype view, phenotype co-occurrence can be modeled to reveal patterns across patients. Hence, patients are encoded from these two distinct views. To mitigate the influence of noise and irrelevant information in biological data, we devise the cross-view contrastive knowledge distillation that distills insights from the biological view to enhance phenotype imputation. Phenotype imputation with the proposed model demonstrates superior performance over state-of-the-art models on the real-world biomedical database.

## 1 Introduction

Clinical records, serving as a critical resource for understanding disease patterns and patient outcomes, are valuable for observational studies. However, its collection can be biased or incomplete due to the limits on infrastructures and expertise, the inconsistency in data types across healthcare systems, and the variability in patient cohorts, etc [3, 15]. For instance, it is recognized that patients with dementia and its related conditions can have under-documented phenotypes [36], probably resulting from a lack of clear symptoms early on or ignorance of related diseases. The issue of missing or incomplete phenotypic data is pervasive and can lead to biased results in medical research and suboptimal patient care [21]. In light of this, *phenotype imputation* is essential to ensure a more holistic and precise medical evaluation, thereby optimizing patient care and enhancing the validity of clinical studies.

Traditional imputation methods [11, 2] rely on informative statistical characteristics of the clinical data to infer the missing phenotypes, yet often neglect the broad, interconnected nature of clinical data with multi-modal biological information such as proteomics and metabolomics, while the latter might provide deeper insights into the patient's health status. The growing development of extensive biobanks [4, 33], collecting various biological and lifestyle data alongside traditional clinical records, unlocks a potential to address incomplete phenotypic data in clinical records. By leveraging multi-modal biological data as external information, as shown in Figure 1, the associations

---

[*]Corresponding Author

38th Conference on Neural Information Processing Systems (NeurIPS 2024).

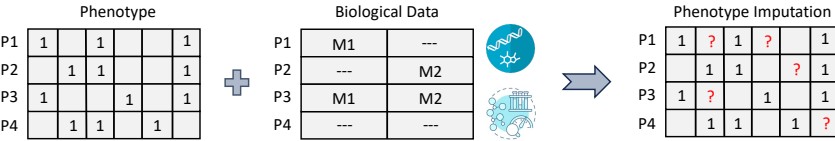

**Figure 1:** Phenotype imputation with multi-modal biological data. "M1" denotes Modality 1, and "M2" represents Modality 2. "—" refers to the missing modality and the red question mark refers to the phenotype that needs to be imputed.

between biological observations and clinical phenotypes might improve the inference of incomplete phenotypes.

However, leveraging multi-modal information for phenotype imputation remains a complex challenge in two folds: 1) The heterogeneity of multi-modal data typically results in significant variances from clinical data, as it includes different data types and characteristics. For instance, continuous variables in proteomics may exhibit different patterns and correlations with a patient's health status compared to discrete phenotype data. Multi-modal biological data often contain measurement noise and irrelevant information unrelated to phenotypic observations, which hinders accurate phenotype imputation. Furthermore, biological data are frequently missing for many individuals due to the labor-intensive and costly nature of data collection.

Despite the compelling need to leverage multi-modal data, the challenges outlined above have posed significant obstacles to developing an effective approach for phenotype imputation. In recent years, graphs have gained traction as a powerful tool for modeling complex data and capturing relationships between real-world entities. Representing patients and phenotypes within a graph structure and imputing missing phenotypes using Graph Neural Networks (GNNs) offers a promising path forward. Biological data could, in principle, be incorporated as patient attributes and propagated through the graph. However, the joint modeling conflicts with the heterogeneity between biological and phenotypic data, as each encapsulates distinct rationales for unveiling patient-specific health conditions. First, *from a statistical and collaborative view*, the patient-phenotype graph connecting patients and their phenotypes reflects phenotype co-occurrence patterns across all patients' interactions. These co-occurrence patterns indicate an underlying principle in imputation: if phenotype $x$ and $y$ are frequently co-diagnosed, it is sensible to impute $y$ for a patient once $x$ is observed. Second, *from a biological view*, a patient's biological data reveals their fine-grained health status. This highlights another rationale for imputation: understanding the detailed health conditions from biological data can guide the imputation of phenotypes that correspond to similar biological health status. Therefore, in this paper, we propose a view decoupling approach to segregate the modeling of biological data from phenotypic data, thereby fully utilizing the information from both sources.

To model the correlation between patients and phenotypes, one can construct and encode a bipartite graph. Nevertheless, the use of biological data is not a straightforward task. Biological data is characteristically composed of a wide range of variables, including protein concentrations, metabolic profiles, gene expression levels, etc. These variables exhibit high-dimensional and continuous characteristics, making it challenging to model the data effectively. More importantly, the biological conditions of patients uncover major underlying factors that indicate health status. In other words, patients sharing similar underlying biological factors could have similar phenotypes. Identifying these latent factors would facilitate the effective characterization of patients and their phenotypes.

To tackle these challenges, in this paper, we propose a novel framework **MPI**, aiming to harness the **M**ultimodal data for **P**henotype **I**mputation. First, to identify the latent biological factors, we propose quantizing the biological data and uncovering the corresponding factors using Residual Quantization. Then, the obtained factors in conjunction with the patients themselves, are utilized to create a graph that models the correlation between patients from a biological view. To decouple views and segregate the modeling of biological data from phenotypic data, the patients and phenotypes are additionally incorporated into another separate graph that depicts the patterns of co-occurrence from the collaborative view. GNNs are then employed to encode both graphs. Second, with the two separate graphs, we aim to leverage the biological information to facilitate the phenotype imputation. However, due to the presence of noise and irrelevant information in biological data, relying solely on biological factors may lead to inaccurate imputation. Thus, we employ a cross-view contrastive knowledge distillation strategy to distill biological knowledge for enhancing phenotype imputation. Within a teacher-student framework, we consider the biological-view GNN as the teacher model and

the collaborative-view GNN as the student model. Rather than replicating the teacher model entirely, the aim is for the student model to glean useful knowledge by receiving partial guidance from the teacher model. The main contributions of this work are summarized as:

1) We propose leveraging multi-modal data to enhance phenotype imputation through view decoupling, thereby segregating the modeling of multi-modal biological data from phenotype data. 2) To enhance the depiction of patient profiling and facilitate the imputation, we propose to uncover the latent biological factors of patients and accordingly model the correlation among the patients based on these factors. 3) To avoid the impact of noise and irrelevant information in biological data, we adopt a novel cross-view contrastive knowledge distillation to subtly leverage information from biological data. 4) Extensive experiments over a real-world biomedical database demonstrate the superiority of our proposed method over state-of-the-art methods.

## 2 Related Work

**Phenotype Imputation.** Phenotype imputation involves predicting missing phenotypic information in clinical electronic health records (EHRs), e.g., diseases and symptoms, generally leveraging various methods ranging from traditional statistical approaches to advanced machine learning techniques. Early research relies on statistical modeling and matrix analysis [41, 40, 10, 1], while deep learning demonstrates effectiveness in modeling more complex dependencies with deep networks [14, 50, 27, 2]. Despite existing efforts to explore the correlations between phenotypes and genotypes [2], multi-modal biological data is largely overlooked in EHR analysis. Our approach differentiates itself by utilizing multi-modal biological data to enhance phenotype imputation in EHRs.

**Graph Neural Networks in Biomedicine.** Graph Neural Networks (GNNs) [13, 54] have been employed to model the interconnectivity of either clinical data or biological information. A line of research devises GNN models for EHRs to enhance healthcare representation learning and patient-specific outcomes [9, 35, 20, 28]. By leveraging the entities and connections in EHRs, e.g., diseases, symptoms, and drug interactions, GNNs show effectiveness in producing patient profiles and clinical predictions [23, 26]. Meanwhile, biological studies leverage GNNs to explore biological networks, promote disease mechanism discovery, analyze drug response, etc. For instance, single-cell biology adopts GNNs to analyze cellular heterogeneity, aiming for an improved understanding of cellular functions and interactions [18, 31]. Besides, some work integrates clinical and molecular data to predict adverse drug reaction signals [22], exemplifying the integration of EHRs and biological data for combined healthcare analysis. Our approach leverages biological data to aid phenotype imputation in EHRs by bridging the gap between clinical data and underlying biological mechanisms.

**Multi-modal Representation Learning on EHRs.** Multi-modal learning on EHRs aims to integrate varied modalities in EHRs, e.g., medication records, lab test results, imaging data, and clinical notes, to obtain optimized patient representations [23, 17]. Given the potential unavailability of modalities, research efforts are made to improve model robustness in the face of partially or completely missing modalities. Strategies include imputing the missing modalities, exploring the data generation process, and preserving the structure of observed data [48, 29, 52, 6, 47, 53]. However, existing works primarily explore modalities within EHRs as clinical insights, often overlooking biological knowledge in EHR analysis. Different from existing work, we explore multi-modal biological data with random missingness to enhance phenotype imputation in EHRs, via addressing the heterogeneity and inaccuracy in multi-modal biological data.

## 3 Preliminaries

**Electronic Health Records (EHRs).** Clinical records, integral for encoding patient health information, are commonly digitized into electronic health records (EHRs) and formatted as high-dimensional medical codes. Typically, a clinical record includes a series of clinical entities, such as diagnoses, medications, procedures, laboratory tests, and clinical notes. In this paper, our primary focus is on the phenotypic information within EHRs, which is generally encoded as one-hot vectors, thus indicating the presence or absence of specific medical symptoms or diseases.

**Phenotype.** Define the phenotype data in EHRs for a patient cohort as $\mathbf{X} = \{\mathbf{x}_1, \mathbf{x}_2, \ldots, \mathbf{x}_N\}$, where $N$ represents the total number of patients. Each $\mathbf{x}_i$ encapsulates the phenotypic attributes for patient $i$, represented by medical codes for symptoms and diseases, denoted as $\mathbf{x}_i = \{p_1, p_2, \ldots, p_{|\mathbf{x}_i|}\}$.

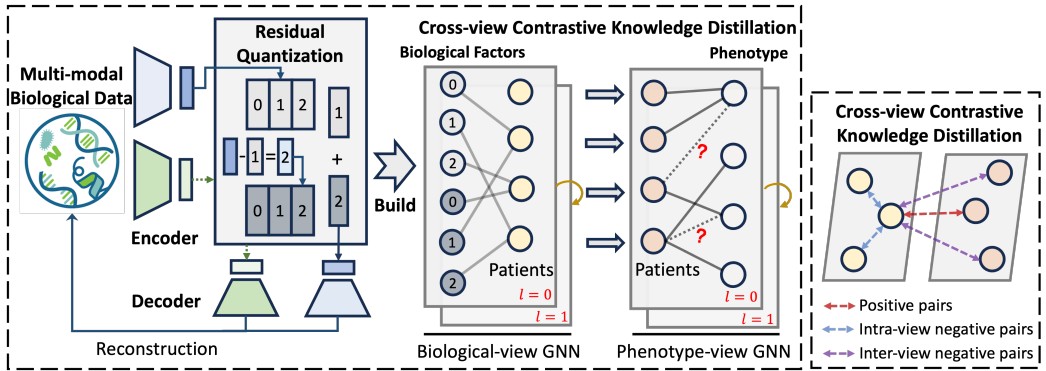

**Figure 2:** An overview of the MPI framework: (1) Residual Quantization quantizes the biological data and uncovers the underlying factors. (2) Biological-view GNN and Phenotype-view GNN are employed to encode the correlation between patients, biological factors, and phenotypes in separate graphs. (3) Cross-view knowledge distillation makes use of learned representations from different views and enhances the imputation.

**Patient Multi-modal Data with Irregular Missingness.** In biological multi-modal datasets, we represent each patient by a collection of data points from various biological modalities, such as genetics, proteomics, or metabolomics. Let $Z$ represent the total number of modalities, then the multi-modal dataset for patients can be expressed as $\mathbf{X}^M = \{\mathbf{x}_1^M, \mathbf{x}_2^M, \ldots, \mathbf{x}_N^M\}$, where $N$ denotes the number of patients. Given the potential for absent modalities, we define the observed multi-modal data for patient $i$ as $\mathbf{x}_i^M = \{\mathbf{x}_i^1, \mathbf{x}_i^2, \ldots, \mathbf{x}_i^m\}$, adhering to the condition $0 \leq m \leq Z$. We focus on the most relaxed setting where the modality missingness is irregular across patients, i.e., random missingness. This randomness persists through the phases of training, validation, and testing, allowing for the possibility that a patient might lack data for any, or in extreme cases, all modalities.

**Phenotype Imputation.** Phenotype imputation aims to address critical gaps in clinical records, where certain medical symptoms, disease attributes, or outcomes are not documented or are incompletely recorded. Given a patient cohort and the incomplete phenotypic data in a clinical dataset, the problem we focus on aims to impute the other possible phenotypes by leveraging available biological multi-modal data. Let $\mathbf{X}$ be the incomplete phenotype data, and $\mathbf{X}^M$ be the biological multi-modal data with irregular missingness, the objective is to design a model that infers the existence of other possible phenotypes. Thereby, a model $\Phi$ is expected to perform $\mathbf{Y} = \Phi(\mathbf{X}, \mathbf{X}^M; \cdot)$ and minimize the discrepancy between the actual phenotype $\tilde{\mathbf{Y}}$ and the imputed phenotype $\mathbf{Y}$. Here $\mathbf{Y}$ and $\hat{\mathbf{Y}}$ denote one-hot vectors. Given the extensive set of phenotypes, measuring discrepancy through classification is impractical. Therefore, we frame the imputation task as a ranking problem, aiming to position the correct phenotype higher than the incorrect ones.

## 4 The Proposed Method

In this section, we introduce the proposed method MPI. As shown in Figure 2, our proposed model includes three components, i.e., biological data quantization, dual-view graph representation learning, and cross-view contrastive knowledge distillation. Next, we describe each component in detail.

### 4.1 Biological Data Quantization

The biological state reveals analogous latent factors among patients. Existing approaches primarily use biological data as features and apply traditional machine learning techniques to encode them, yet they often struggle to disentangle the complex, heterogeneous factors inherent in biological data [44, 12]. The learned representation of patients could be non-robust (e.g., prone to overreact to an irrelevant factor) and hardly explainable. To identify the latent biological factors among patients, we propose quantizing the biological data and uncovering the corresponding factors using residual quantization [24], which employs a multi-level vector quantizer to convert residuals into a series of codes. Specifically, the input $\mathbf{x}^m$ is initially encoded into a latent representation $\mathbf{z}^m := \mathbf{E}(\mathbf{x}^m)$ by an encoder $\mathbf{E}$. At the first level ($d = 0$), the residual is set to $\mathbf{r}_0 := \mathbf{z}^m$. For each level $d$, we define a codebook $C_d := \{\mathbf{e}_k\}_{k=1}^K$ with size $K$. The residual $\mathbf{r}_0$ is quantized by mapping it to the

nearest embedding from the codebook. The index of the closest embedding $\mathbf{e}_{c_0}$ at $d = 0$, which is $c_0 = \arg\min_k \|\mathbf{r}_0 - \mathbf{e}_k\|$, represents the zero-th code. For the next level ($d = 1$), the residual is updated to $\mathbf{r}_1 := \mathbf{r}_0 - \mathbf{e}_{c_0}$. The code for this level is determined by finding the embedding in the first level's codebook that is nearest to $\mathbf{r}_1$. This quantization process is recursively repeated $l$ times, producing a tuple of $l$ codes that constitute the disentangled biological factors. This hierarchical approach approximates the input biological data from coarse to fine granularity. Notably, separate codebooks are used for each of the $l$ levels rather than a single, large codebook. This strategy is preferred as the norm of residuals tends to decrease with increasing levels, facilitating the capture of different granularity levels from the input data.

Upon obtaining the disentangled biological factors $(c_0, \ldots, c_{l-1})$, the quantized representation of $\mathbf{z}^m$ is determined as $\hat{\mathbf{z}}^m := \sum_{d=0}^{l-1} \mathbf{e}_{c_d}$. This quantized vector $\hat{\mathbf{z}}^m$ is subsequently fed into a decoder $\mathbf{D}$, which attempts to reconstruct the input $\mathbf{x}^m$ based on $\hat{\mathbf{x}}^m = \mathbf{D}(\hat{\mathbf{z}}^m)$. The loss function for the residual quantization is defined as follows:

$$\mathcal{L}_{\text{bio}} := \mathcal{L}_{\text{recon}} + \mathcal{L}_{\text{rq}}, \tag{1}$$

where $\mathcal{L}_{\text{recon}} := \|\mathbf{x}^m - \hat{\mathbf{x}}^m\|^2$ and $\mathcal{L}_{\text{rq}} := \sum_{i=0}^{l-1} \|\text{sg}[\mathbf{r}_i] - \mathbf{e}_{c_i}\|^2 + \beta\|\mathbf{r}_i - \text{sg}[\mathbf{e}_{c_i}]\|^2$. Here, $\hat{\mathbf{x}}^m$ represents the decoder's output, and sg denotes the stop-gradient operation [42]. The training of this autoencoder involves simultaneous updating of the quantization codebooks and the parameters of the encoder-decoder. Note that the exclusive autoencoder and quantization codebooks are learned to capture the disentangled biological factors for each modality. For example, a patient's biological data includes two types of modalities, the disentangled biological factors can be represented as $(c_0^1, \ldots, c_{l-1}^1)$ and $(c_0^2, \ldots, c_{l-1}^2)$. We use $\mathcal{C}$ to denote the set of learned biological factors in all codebooks in subsequent sections.

## 4.2 Dual-view Graph Representation Learning

With disentangled biological factors and phenotypes, a patient can be described from two perspectives: a phenotype view and a biological view. To effectively capture the relationship between patients and biological factors and phenotypes, and fully utilize the information from both views, we construct two separate graphs instead of a single patient-centric graph.

**Patient-Phenotype Graph Construction**. From the phenotype view, we construct a patient-phenotype graph, denoted as $\mathcal{G}_p$, to depict the collaborative relationships between phenotypes, specifically focusing on phenotype-phenotype co-occurrences. The construction of $\mathcal{G}_p$ begins with defining a set of phenotypes $\mathcal{P}$ and a set of patients $\mathbf{X}$. Each patient $\mathbf{x} \in \mathbf{X}$ is associated with one or more phenotypes $p \in \mathcal{P}$. An edge is created between a patient node and a phenotype node if the patient exhibits that phenotype. By linking patients to their respective phenotypes, $\mathcal{G}_p$ captures the complex interactions and shared occurrences of different phenotypes across the patient cohort, and provides a comprehensive view of how different phenotypes interact within the patient population.

**Patient-Factor Graph Construction**. From the biological view, we first construct a patient-factor graph, denoted as $\mathcal{G}_f$, to explore the biology-level correlation between patients. Specifically, the graph $\mathcal{G}_f$ is constructed using the same set of patients $\mathbf{X}$ and disentangled biological factors $\mathcal{C}$ from learned codebooks as the set of nodes. To connect patients and factors, we build edges between each patient $\mathbf{x}$ and their corresponding factors $(c_0, \ldots, c_{l-1})$. This patient-factor graph $\mathcal{G}_f$ reveals patient correlations through shared factors, offering a distinct approach to characterizing patients.

With the constructed graphs $\mathcal{G}_f$ and $\mathcal{G}_p$, we denote the adjacency matrices of $\mathcal{G}_f$ and $\mathcal{G}_p$ as $\mathbf{A}_f$ and $\mathbf{A}_p$, respectively. To capture the structural information of the graphs $\mathcal{G}_f$ and $\mathcal{G}_p$ and learn the representation of patients, phenotypes, and biological factors, we utilize basic Graph Convolutional Networks (GCNs) as the graph encoder. Taking $\mathcal{G}_p$ as an example, the phenotype-view graph encoder for $\mathcal{G}_p$ works by:

$$\mathbf{H}_p^{(l+1)} = \sigma\left(\hat{\mathbf{A}}_p \mathbf{H}_p^{(l)} \mathbf{W}_p^{(l)}\right), \tag{2}$$

where $\mathbf{H}_p^{(0)} = \mathbf{F}_p$ represents the initial input features, to be more specific, for patients and phenotypes, the input features are randomly initialized. In contrast, for biological factors, the input features are initialized using the corresponding code embedding of factors. And $\mathbf{H}_p^{(l)}$ denotes the node representations at the $l$-th layer. The matrix $\hat{\mathbf{A}}_p = \hat{\mathbf{D}}_p^{-1/2} \tilde{\mathbf{A}}_p \hat{\mathbf{D}}_p^{-1/2}$ is the symmetrically normalized adjacency matrix, with $\hat{\mathbf{D}}_p \in \mathbb{R}^{N \times N}$ being the degree matrix of $\tilde{\mathbf{A}}_p = \mathbf{A}_p + \mathbf{I}_N$, where $\mathbf{I}_N$ is

the identity matrix. Similarly, the representation $\mathbf{H}_f^{(l)}$ can be learned from the graph $\mathcal{G}_f$ using the biological-view graph encoder.

To optimize both graph encoders and to effectively differentiate between the positive and negative edges in graphs, we define a margin-based ranking loss for graph $\mathcal{G}_p$ as follows:

$$\mathcal{L}_p = \sum_{(i,j)\in\mathcal{E}_p} \sum_{(i,k)\in\mathcal{N}_p} \max(0, \gamma - f(i,j) + f(i,k)), \qquad (3)$$

where $\gamma$ is the margin hyperparameter, $(i,j) \in \mathcal{E}_p$ denotes the set of positive edges in graph $\mathcal{G}_p$, and $(i,k) \in \mathcal{N}_p$ denotes the set of negative edges and $(i,k)$ does not present in $\mathcal{G}_p$. $f(,)$ is a multi-layer perceptron (MLP) that takes node embeddings as inputs and outputs the similarity score between two node embeddings. We use the same loss function to update the biological-view graph encoder of graph $\mathcal{G}_f$ and denote the loss as $\mathcal{L}_f$.

## 4.3 Cross-view Contrastive Knowledge Distillation

Due to the noisy and irrelevant information in the biological data that could mislead the phenotype imputation, the learning from the biological view and the learning from the phenotype view are separative and we propose a cross-view contrastive knowledge distillation strategy to subtly leverage the biological knowledge to facilitate the phenotype imputation. Following the teacher-student framework [19, 8, 39], we regard the biological-view graph encoder as the teacher model and the phenotype-view graph encoder as the student model. Since the teacher model cannot provide the completely precise knowledge to represent patients [34], instead of fully imitating the behavior of the teacher model, the student model is expected to extract the beneficial knowledge only incorporating partial supervision from the teacher model. Specifically, with the patient representation $\mathbf{H}_f$ learned from biological-view graph $\mathcal{G}_f$ and patient representation $\mathbf{H}_p$ learned from the collaborative-view graph $\mathcal{G}_p$, we propose cross-view contrastive knowledge distillation to distill useful knowledge from the biological-view graph encoder. This approach leverages view-specific embeddings, represented as $\mathbf{h}_f^i$ from the biological view and $\mathbf{h}_p^i$ from the phenotype view for patient $i$. Our objective is to align these embeddings into a shared space, facilitating discriminative representation learning through contrastive loss. Initially, embeddings are processed through a transformation with hidden layers to project them into the desired space as $\mathbf{h}_f^i = \sigma\left(\mathbf{W}^{(2)}\sigma\left(\mathbf{W}^{(1)}\mathbf{h}_f^i + \mathbf{b}^{(1)}\right) + \mathbf{b}^{(2)}\right)$ where $\mathbf{W}^{(1)}$ and $\mathbf{W}^{(2)}$ are the trainable weight matrices, $\mathbf{b}^{(1)}$ and $\mathbf{b}^{(2)}$ are the bias terms, and $\sigma$ represents the ELU activation function. $\mathbf{h}_p^i$ can also be processed using the same transformation.

We then define positive and negative samples to compute the contrastive loss. Embeddings of the same patient form positive samples from two different views, while negative samples consist of embeddings from different patients. Specifically, for a given patient $i$, the positive sample pair is $(\mathbf{h}_f^i, \mathbf{h}_p^i)$, and negative samples include both intra-view and inter-view pairs. The contrastive knowledge distillation loss is formulated as follows:

$$\mathcal{L}_{\text{CKD}} = -\log \frac{e^{s(\mathbf{h}_f^i, \mathbf{h}_p^i)/\tau}}{e^{s(\mathbf{h}_f^i, \mathbf{h}_p^i)/\tau} + \sum_{k\neq i}\left(e^{s(\mathbf{h}_f^i, \mathbf{h}_f^k)/\tau}\right) + \sum_{k\neq i}\left(e^{s(\mathbf{h}_f^i, \mathbf{h}_p^k)/\tau}\right)} \qquad (4)$$

where $s(\cdot, \cdot)$ denotes the cosine similarity, and $\tau$ is a temperature parameter. This loss function incorporates negative samples from both intra-view and inter-view sources, ensuring a comprehensive learning process. By applying this cross-view contrastive optimization, our model effectively captures the intricate relationships within both the biological and collaborative views, leading to robust representations of the patients. Since the biological knowledge is distilled from the biological-view graph encoder to enhance the phenotype-view graph encoder, the loss function for $\mathcal{G}_p$ to optimize the phenotype-view graph encoder is updated to $\hat{\mathcal{L}}_p = \mathcal{L}_p + \alpha\mathcal{L}_{\text{CKD}}$ where $\alpha$ is a tradeoff parameter.

## 4.4 Optimization

In optimization, residual quantization involves the pretraining of autoencoders for biological data, and the quantization codebooks using loss function $\mathcal{L}_{\text{bio}}$ to learn the disentangled biological factors and their corresponding factor embeddings. Subsequently, we utilize an iterative optimization strategy to optimize the biological-view graph encoder using $\mathcal{L}_f$ and phenotype-view graph encoder using $\hat{\mathcal{L}}_p$.

**Table 1:** Dataset Statistics

| Dataset | Unique Items# | Interactions# | Sparsity/Missing Rates |
|---------|---------------|---------------|------------------------|
| **Patient** | 15,093 | - | - |
| **Phenotype** | 1,109 | 380,239 | 97.73% |
| **Proteomics** | 2,923 | 1,483 | 90.2% |
| **Metabolomics** | 251 | 7,513 | 50.3% |

Specifically, we leverage the patient representation learned from the biological view as the teacher signal and optimize the phenotype-view graph encoder through contrastive knowledge distillation following loss function $\hat{\mathcal{L}}_p$. The process is iterated until both graph encoders converge. During the evaluation phase, we employ the patient representation learned from the phenotype-view graph encoder and evaluate a positive testing phenotype along with a set of candidate negative phenotypes to assess performance. The pseudocode of MPI training procedure is described in Algorithm 1.

# 5 Real-World Experiments

## 5.1 Experimental Setup

**Dataset.** We evaluate MPI and baseline approaches using the UK Biobank [4], a comprehensive biomedical database and research resource collecting extensive biological samples and clinical EHRs. We focus on phenotype imputation for populations suffering from chronic diseases and thus extract a cohort of patients diagnosed with Alzheimer's disease and related dementia. Specifically, we leverage the EHRs from inpatient and primary care to obtain phenotypic data before disease onset after preprocessing and transformation. Besides, we utilize biological data across two modalities: proteomics, measuring levels of roughly 3,000 proteins; and metabolomics, testing around 250 metabolic biomarkers. The biological data is preprocessed following common practice [7, 55]. We observe significant modality missingness at random: approximately 90% in proteomics and 50% in metabolomics. Table 1 shows the statistics of the dataset, with dataset details and preprocessing methods described in the Appendix A.1.

**Baselines.** We compare the proposed model to baselines across three categories: (1) modality imputation methods, including **CMAE** [32] and **SMIL** [30]; (2) graph neural networks comprising **GraphSage** [16] and **GIN** [49], which utilize multi-modal biological information as patient features; (3) multi-modal models on EHRs that handle missingness, consisting of **M3Care** [53], **GRAPE** [51] and **MUSE** [47]. Note that all these methods primarily focus on patient classification tasks and rely on supervision signals from patient labels. We adapt their training objectives to suit our problem setting and evaluate the baselines on the same testing data for a fair comparison. Additional details on the baselines are provided in Appendix A.2.

**Experimental Settings.** We implement MPI with PyTorch and run it on an NVIDIA RTX A6000 GPU. To implement MPI, a two-layer GCN is utilized for each decomposed view with 128 and 64 hidden units respectively. It's worth noting that our focus is not on the complexity of the GNN itself; we use GCN as the foundational backbone model, which can be substituted with any advanced GNNs as needed. Besides, the quantization of proteomics and metabolomics is conducted with respective autoencoders including a two-layer encoder and one-layer decoder, with a hidden size of 32 units. To determine the trade-off weight for knowledge distillation, we choose 0.1 after a grid search in {0.01, 0.1, 1, 5, 10}. The margin hyperparameter $\gamma$ is determined as 3 through a search in {1, 3, 5,10}. The model is trained with Adam optimizer and evaluated at every epoch with an early-stopping strategy at patience of 40 per the validation set performance. Baselines including Graphsage and GIN utilize the same hidden sizes as MPI. CMAE and SMIL first conduct feature imputation for the missing modalities, afterwards an MLP model is conducted with the imputed features for our ranking objectives. As M3Care, Grape, and MUSE build graphs for patients and EHR modalities, we use their published implementations and conduct adaptations to suit our problem setting. Thus, we build the connections between patients and multi-modal modalities and meanwhile incorporate patient phenotype connections for a fair comparison. Baseline hyperparameters are determined by parameter search. Besides, the model learning rate is selected from {0.01, 0.001, 0.0005} for MPI and all baseline models.

**Table 2:** Performance comparison for different models on varying dataset proportions.

| % | Metric | CMAE | SMIL | GraphSage | GIN | GRAPE | M3Care | MUSE | MPI |
|---|---|---|---|---|---|---|---|---|---|
| 30% | H@10 | $25.81^{\pm 0.14}$ | $26.12^{\pm 0.25}$ | $24.96^{\pm 0.77}$ | $25.36^{\pm 0.66}$ | $25.60^{\pm 0.64}$ | $\underline{26.23^{\pm 0.56}}$ | $24.24^{\pm 0.32}$ | $\mathbf{28.87^{\pm 0.04}}$ |
| | H@20 | $41.66^{\pm 0.42}$ | $41.08^{\pm 0.53}$ | $40.61^{\pm 0.47}$ | $41.51^{\pm 0.84}$ | $41.41^{\pm 0.73}$ | $\underline{41.90^{\pm 0.65}}$ | $40.89^{\pm 0.58}$ | $\mathbf{44.45^{\pm 0.44}}$ |
| | H@50 | $68.81^{\pm 0.16}$ | $68.23^{\pm 0.21}$ | $67.28^{\pm 0.20}$ | $69.02^{\pm 0.68}$ | $68.45^{\pm 0.25}$ | $68.71^{\pm 0.34}$ | $67.90^{\pm 0.24}$ | $\mathbf{70.24^{\pm 0.15}}$ |
| | MRR | $11.51^{\pm 0.13}$ | $11.46^{\pm 0.32}$ | $11.23^{\pm 0.52}$ | $11.50^{\pm 0.30}$ | $11.33^{\pm 0.27}$ | $\underline{11.87^{\pm 0.25}}$ | $11.06^{\pm 0.35}$ | $\mathbf{13.22^{\pm 0.17}}$ |
| 50% | H@10 | $26.33^{\pm 0.28}$ | $26.57^{\pm 0.28}$ | $28.59^{\pm 0.23}$ | $\underline{29.35^{\pm 0.39}}$ | $28.83^{\pm 0.47}$ | $27.43^{\pm 0.32}$ | $28.86^{\pm 0.43}$ | $\mathbf{31.28^{\pm 0.32}}$ |
| | H@20 | $42.34^{\pm 0.35}$ | $42.68^{\pm 0.42}$ | $44.51^{\pm 0.40}$ | $\underline{45.58^{\pm 0.47}}$ | $45.20^{\pm 0.38}$ | $44.66^{\pm 0.26}$ | $44.87^{\pm 0.36}$ | $\mathbf{47.55^{\pm 0.31}}$ |
| | H@50 | $69.28^{\pm 0.51}$ | $69.35^{\pm 0.22}$ | $70.92^{\pm 0.20}$ | $\underline{71.82^{\pm 0.34}}$ | $70.74^{\pm 0.34}$ | $70.28^{\pm 0.51}$ | $70.77^{\pm 0.40}$ | $\mathbf{72.99^{\pm 0.14}}$ |
| | MRR | $11.99^{\pm 0.04}$ | $12.09^{\pm 0.19}$ | $13.30^{\pm 0.23}$ | $\underline{13.77^{\pm 0.28}}$ | $13.14^{\pm 0.29}$ | $12.64^{\pm 0.21}$ | $13.41^{\pm 0.31}$ | $\mathbf{14.83^{\pm 0.15}}$ |
| 70% | H@10 | $27.40^{\pm 0.55}$ | $28.24^{\pm 0.35}$ | $32.35^{\pm 0.18}$ | $\underline{33.13^{\pm 0.41}}$ | $30.51^{\pm 0.63}$ | $30.68^{\pm 0.49}$ | $32.42^{\pm 0.73}$ | $\mathbf{35.68^{\pm 0.56}}$ |
| | H@20 | $43.50^{\pm 0.37}$ | $44.54^{\pm 0.31}$ | $48.12^{\pm 0.25}$ | $\underline{49.12^{\pm 0.35}}$ | $47.07^{\pm 0.59}$ | $46.53^{\pm 0.35}$ | $48.38^{\pm 0.59}$ | $\mathbf{51.59^{\pm 0.48}}$ |
| | H@50 | $69.93^{\pm 0.31}$ | $70.28^{\pm 0.26}$ | $73.10^{\pm 0.31}$ | $\underline{73.18^{\pm 0.40}}$ | $72.64^{\pm 0.53}$ | $71.73^{\pm 0.47}$ | $73.04^{\pm 0.61}$ | $\mathbf{75.82^{\pm 0.34}}$ |
| | MRR | $12.48^{\pm 0.28}$ | $13.36^{\pm 0.18}$ | $15.59^{\pm 0.36}$ | $\underline{15.75^{\pm 0.22}}$ | $14.04^{\pm 0.45}$ | $13.75^{\pm 0.26}$ | $15.19^{\pm 0.58}$ | $\mathbf{17.44^{\pm 0.41}}$ |
| 90% | H@10 | $27.90^{\pm 0.26}$ | $29.03^{\pm 0.27}$ | $\underline{35.48^{\pm 0.30}}$ | $35.41^{\pm 0.35}$ | $31.61^{\pm 0.25}$ | $32.55^{\pm 0.33}$ | $33.73^{\pm 0.34}$ | $\mathbf{37.74^{\pm 0.32}}$ |
| | H@20 | $44.10^{\pm 0.31}$ | $46.15^{\pm 0.42}$ | $\underline{51.47^{\pm 0.32}}$ | $51.36^{\pm 0.25}$ | $48.86^{\pm 0.36}$ | $48.24^{\pm 0.55}$ | $49.84^{\pm 0.37}$ | $\mathbf{53.77^{\pm 0.46}}$ |
| | H@50 | $70.42^{\pm 0.13}$ | $71.58^{\pm 0.15}$ | $\underline{75.45^{\pm 0.15}}$ | $74.95^{\pm 0.55}$ | $73.40^{\pm 0.32}$ | $73.38^{\pm 0.28}$ | $74.63^{\pm 0.30}$ | $\mathbf{77.44^{\pm 0.25}}$ |
| | MRR | $12.77^{\pm 0.16}$ | $13.43^{\pm 0.09}$ | $\underline{17.36^{\pm 0.06}}$ | $17.33^{\pm 0.11}$ | $15.16^{\pm 0.18}$ | $15.06^{\pm 0.22}$ | $16.34^{\pm 0.26}$ | $\mathbf{18.63^{\pm 0.22}}$ |
| 100% | H@10 | $28.02^{\pm 0.34}$ | $29.87^{\pm 0.43}$ | $\underline{36.64^{\pm 0.29}}$ | $36.61^{\pm 0.07}$ | $32.70^{\pm 0.21}$ | $33.54^{\pm 0.34}$ | $34.92^{\pm 0.31}$ | $\mathbf{38.74^{\pm 0.27}}$ |
| | H@20 | $44.29^{\pm 0.29}$ | $46.53^{\pm 0.32}$ | $\underline{53.01^{\pm 0.42}}$ | $52.69^{\pm 0.38}$ | $49.68^{\pm 0.44}$ | $50.32^{\pm 0.42}$ | $50.94^{\pm 0.45}$ | $\mathbf{55.10^{\pm 0.31}}$ |
| | H@50 | $70.64^{\pm 0.25}$ | $72.01^{\pm 0.18}$ | $\underline{76.58^{\pm 0.11}}$ | $76.32^{\pm 0.16}$ | $74.27^{\pm 0.27}$ | $73.18^{\pm 0.45}$ | $75.62^{\pm 0.26}$ | $\mathbf{78.42^{\pm 0.20}}$ |
| | MRR | $12.88^{\pm 0.20}$ | $14.27^{\pm 0.24}$ | $\underline{17.99^{\pm 0.23}}$ | $17.94^{\pm 0.22}$ | $15.63^{\pm 0.30}$ | $15.28^{\pm 0.32}$ | $16.61^{\pm 0.29}$ | $\mathbf{19.28^{\pm 0.19}}$ |

**Table 3:** Ablation study of variants comparison on 30% and 100% of the dataset.

| Variants | | | | 100% | | | | 30% | | | |
|---|---|---|---|---|---|---|---|---|---|---|---|
| | Prote. | Metabol. | CKD | Hits@10 | Hits@20 | Hits@50 | MRR | Hits@10 | Hits@20 | Hits@50 | MRR |
| V1 | | | | 36.49 | 52.75 | 76.53 | 17.73 | 26.39 | 42.21 | 68.91 | 11.98 |
| V2 | ✓ | | ✓ | 38.13 | 54.70 | 77.48 | 19.08 | 27.94 | 44.26 | 69.80 | 13.09 |
| V3 | | ✓ | ✓ | 38.31 | 54.75 | 78.01 | 19.02 | 28.21 | 44.36 | 69.87 | 13.11 |
| V4 | ✓ | ✓ | | 37.68 | 53.99 | 77.95 | 18.40 | 27.89 | 43.79 | 69.07 | 12.32 |
| MPI | ✓ | ✓ | ✓ | 38.74 | 55.10 | 78.41 | 19.27 | 28.87 | 44.45 | 70.24 | 13.22 |

**Evaluation Protocol.** The discussion on the evaluation protocol can be found in the Appendix A.3.

## 5.2 Experimental Results

**Performance Comparison.** Table 2 presents the performance of the MPI and baseline models trained with different proportions of the dataset. The best results are highlighted in **bold**, while the top baseline scores are underlined. The baselines based on imputation, including CMAE and SMIL, exhibit inferior performance. We attribute this to their reliance on modeling transformations from the hidden space to reconstruct the input features. The imputed data can be inaccurate due to the high dimensionality of the multi-modal data and the severity of missingness. GraphSage and GIN achieve competitive performance compared to both imputation-based models and the multi-modal learning approaches that explicitly handle missing data. The graph-based multi-modal models outperform GNNs in some cases; however, they are sometimes inferior to applying naive integration of clinical and biological data in naive GNNs. This may be due to the complexity and conflict between clinical and biological views. For example, GRAPE, which uses each feature dimension as a node, is not suitable for high-dimensional feature imputation. Additionally, M3Care computes patient similarity for each modality separately, thereby failing to explore cross-modality correlations. MUSE connects patients with modalities while representing each modality type as a node, possibly introducing dense and noisy edges. In contrast, MPI demonstrates improvements across all settings, verifying its capability to handle heterogeneity and noise through a decoupled view.

**Ablation Study.** To validate the effectiveness of MPI and gain deeper insight into the contributions of each component in the proposed approach, we conduct ablation studies by comparing the following variants with the original MPI: (1) V1, which does not utilize the biological data and only model the correlation of patients and phenotypes. (2) V2, which only uses proteomics data and contrastive knowledge distillation. (3) V3, which solely leverages metabolomics data and contrastive knowledge

distillation. (4) V4, which organizes biological factors, patients, and phenotypes in a single graph and does not require contrastive knowledge distillation. The results on 30% and 100% of the UK biobank dataset are summarized in Table 3. First, we observe that variant V1 is outperformed by both V2 and V3. This performance disparity arises since V2 and V3 effectively model the biological data and distill beneficial knowledge, thus enhancing phenotype imputation through knowledge distillation. Second, V4 is inferior to the proposed model MPI. This demonstrates that modeling biological data and phenotype data in separate graphs yields better performance compared to a single graph model. The likely reason for this is that multi-modal biological data often contain measurement inaccuracies and irrelevant information, which can impede accurate phenotype imputation Third, we observe that V3 exhibits superior performance compared to V2. We attribute this to the higher sparsity ratio of proteomics data relative to metabolomics data. The severe missing data issue in proteomics likely affects the performance of imputation. Lastly, compared to all variants, MPI demonstrates the best performance, highlighting the effectiveness of the proposed method.

**The Impact of Codebook Settings.** To analyze the impact of codebook settings on imputation performance, we varied the number and sizes of the codebooks and the results for the entire dataset are presented in Figure 3. First, as shown in Figure 3(left), MPI achieves optimal performance with three codebooks. A smaller number of codebooks, such as one or two, may fail to capture sufficient fine-grained information from the biological data.

Conversely, larger codebooks might introduce additional underlying factors due to finer granularity, which could reduce their discriminative power for patient profiling. Second, Figure 3(right) illustrates that the performance of MPI varies with changes in codebook sizes. The optimal codebook sizes for proteomics and metabolomics are 64 and 96, respectively. Smaller codebook sizes may fail to capture underlying biological factors, resulting in insufficient information for patient profiling. Conversely, larger codebook sizes might lead to certain codes being underutilized, which can hinder the overall optimization of the codebook.

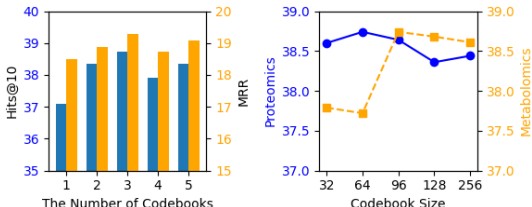

**Figure 3:** (Left) Results for varying the number of codebooks while keeping the codebook size fixed. (Right) Performance variation with changes in the codebook sizes while keeping a fixed number of codebooks.

**Sensitivity to Tradeoff Parameter.** Figure 4 illustrates the impact of varying tradeoff parameters on the performance of MPI, evaluated on 30% and 100% of the dataset. The tradeoff parameter mediates between the contrastive knowledge distillation loss and the graph representation loss. The results indicate that MPI achieves optimal performance with a tradeoff parameter of 0.01.

Notably, when the tradeoff parameter is set to 0, the imputation performance largely declines. This is due to the disabling of knowledge distillation, which prevents the model from leveraging biological knowledge. Conversely, as the tradeoff parameter increases to a high value, the model's performance diminishes. The model might overly depends on biological knowledge and neglects the information from the collaborative view, leading to suboptimal outcomes.

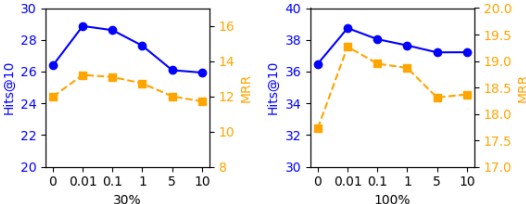

**Figure 4:** Effect of tradeoff parameter for MPI on 30% (left) and 100% (right) of the dataset.

# 6 Conclusion

In conclusion, this work introduces a novel framework that leverages multi-modal data to enhance phenotype imputation, aiming for a more comprehensive medical evaluation. The proposed approach involves uncovering latent biological factors to enhance patient profiling and modeling correlations based on these factors. To mitigate the impact of noise and irrelevant information in biological data, we employ a cross-view contrastive knowledge distillation technique. Extensive experiments on a large-scale biomedical database demonstrate that our proposed method outperforms existing state-of-the-art approaches, showcasing its effectiveness and potential for improving biomedical data analysis and patient care.

## Acknowledgments and Disclosure of Funding

The data utilized in this study were obtained through the UK Biobank Application Number 98304. The authors express their gratitude to all UK Biobank participants for their generous contribution of time to the study. This research is supported by NSF 2212175, NIH RF1AG084178, R01AG076448, R01AG080624, R01AG076234, R01AG080991 and RF1AG072449.

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

# A    Appendix

## A.1    Database

We conduct the experimental evaluation for the proposed model and the baselines on the UK Biobank database [4], which is a lasting endeavor for biomedical research. The UK Biobank recruited half a million participants between 2006 and 2010, maintaining a collection of long-term EHRs from distributed health assessment centers with de-identified lifestyle and health information while retaining biological samples for detailed biological analyses. Chronic diseases frequently exhibit missing phenotypes due to mild or nonspecific initial symptoms. Routine data collection processes might overlook these subtle signs until more pronounced symptoms emerge. This can be particularly challenging in the context of neurodegenerative diseases like Alzheimer's Disease and Related Dementias (ADRD), where early detection is crucial for timely intervention and management. Research on ADRD particularly emphasizes early detection and intervention, which aligns well with the research goals of identifying historical phenotypes. Meanwhile, as one of the most common neurodegenerative diseases, ADRD cohorts might include a wide range of phenotypic expressions and stages of disease. This diversity is crucial for studying the full spectrum of phenotype presentation and identifying underlying missing signs. Therefore, we focus on phenotype imputation for populations suffering from chronic diseases and extracting a specific cohort of patients diagnosed with Alzheimer's disease and related dementia.

To build this cohort, we leverage the HESIN inpatient EHRs and the primary care EHRs from the UK Biobank. As the EHRs are collected from distributed places and organizations, the medical codings vary across different systems. Thus we standardize their variously formatted diagnosis records into uniform ICD codes [5], and filter for patients with ADRD-related ICD codes, following methodologies employed in related research [25]. We further refine the cohort by removing patients whose ADRD onset occurred before or within one year of their biological sample collection, ensuring that the biological information and EHR data used in our analysis reflect the preclinical states of the disease and minimizing confounding factors post-diagnosis. For the extracted cohort, we eliminate any EHRs recorded after the ADRD onset dates, and preprocess the EHRs by converting recorded diagnoses and symptoms into distinct phenotypes [46]. We filter out phenotypes with an occurrence of less than 20 while our cohort population reaches around 15000. The small occurrence ($0.06\%$) reflects the less practical value in this work of imputing these phenotypes, and meanwhile, their rarity often introduces noise rather than providing valuable insights. Besides, there are a few phenotypes with quite high frequency (e.g., hypertension). Since ADRD generally focuses on the elderly population, the widespread prevalence typically indicates a low specificity and can be regarded as possible confounders due to aging. These phenotypes may dominate the dataset, potentially obscuring other important associations, whereas focusing on moderately prevalent phenotypes could uncover more subtle associations.

Beyond EHR data, proteomic analysis has been conducted on blood plasma samples from over 56,000 UK Biobank participants. Enabled by Olink's Proximity Extension Assay (PEA) [45], this analysis measured the abundance of nearly 3,000 circulating proteins. Additionally, the UK Biobank measures around 250 metabolic biomarkers in EDTA plasma samples from approximately 280,000 participants. These biomarkers span multiple metabolic pathways, including lipoprotein lipids, fatty acids, and low-molecular-weight metabolites. Since biological processes could begin years before the onset of clinical symptoms, proteomics and metabolomics which comprise the end-product of genes, transcripts, and protein regulations, offer insights into identifying alterations in multiple biochemical processes and the risk of ADRD among cognitively healthy adults [55, 37]. We leverage biological data across the two modalities of proteomics and metabolomics. Specifically, proteomics data are provided as Normalized Protein eXpression (NPX) values, obtained after UK Biobank preprocessing, which includes median centering normalization between plates and $\log$ transformation. We used these NPX values directly as the encoder input without further processing [7]. For metabolomics, we applied a natural logarithmic transformation ($\ln(x + 1)$) to all metabolite values, followed by Z-transformation [55]. Owing to the resource-intensive nature of these tests and the random unavailability for certain patients, we observe significant modality missingness at random: approximately 90% in proteomics and 50% in metabolomics.

## A.2 Addiontal Details on Baselines

We compare MPI to baselines as follows.

- **CMAE** [32] employs a cross-modality auto-encoder to address missing modalities. Initially, a subset of patients who have complete modalities is sampled where CMAE is trained to reconstruct a purposely masked-out modality. After training, the CMAE model is used to fill in missing modalities for all patients. We use the imputed modality information to perform downstream phenotype imputation via ranking objective.

- **SMIL** [30] integrates Bayesian meta-learning techniques to modify the latent feature space, enabling embeddings with missing modalities to closely resemble those with complete modalities. SMIL estimates the missing modality using a weighted sum of modality priors based on the complete modalities. We adopt the same strategy as CMAE to use SMIL to perform downstream phenotype imputation.

- **GraphSage** [16] is evaluated by learning on the bipartite graph built from EHRs to form the patient-phenotype graph. The built graph will directly leverage the multi-modal biological information as the node features for the patient nodes, where missing biological information is represented as zeros vectors. This baseline serves as a naive combination of clinical data and biological data via joint modeling.

- **GIN** [49] follows the same setup with Graphsage when evaluated. We use the ranking loss to train the Graphsage and GIN baselines.

- **GRAPE** [51] infers missing features by building a bipartite graph to include patients and individual feature dimensions as the graph nodes. The value of the feature is regarded as an edge attribute, where the target is to predict the value assigned to each edge. In this work, around 3000 proteomic features and 250 metabolomics features are included in the graph alongside the patient nodes. We meanwhile include the phenotype nodes and their connections with patients for a fair comparison.

- **M3Care** [53] aims for patient representation learning. It calculates patient similarity within each modality and constructs a similarity graph for each modality respectively. Afterward, overall patient similarities by averaging the similarities from each modality are utilized to model cross-patient interactions by GNNs. The embeddings for each patient across different modalities are then aggregated using a Transformer head. We leverage the learned patient representations in the same way with CMAE and SMIL.

- **MUSE** [47] models the patient-modality relationship in a bipartite graph, where patients and modalities constitute the graph nodes, and modality features serve as edges between them. MUSE applies a Siamese GNN on the bipartite graph and its augmented graph that is obtained via random edge dropout. We also incorporate phenotype nodes in MUSE in a similar manner as in GRAPE to address our specific problem and ensure a fair evaluation.

## A.3 Evaluation Protocol

We randomly hold out 10% of the patient-phenotype interactions as the testing set and train a model on the remaining interactions following previous works [43, 38, 56]. From the training set, we randomly select 10% of the interactions as the validation set to monitor the training process and help early stop. For each observed patient-phenotype interaction, we treat it as one positive pair, while negative instances are sampled from negative phenotypes with which the patient has no interactions. Upon training our model, we generate personalized ranking lists for each patient in the test set, where these lists rank the phenotypes not observed for each patient during training. To evaluate our model's effectiveness, we assess performance using Hit Ratio at specific thresholds (Hit@10, Hit@20, Hit@50) and Mean Reciprocal Rank (MRR). Hit Ratio, as a recall-based metric, measures whether the test phenotype appears within the top-$K$ list. The MRR is position-sensitive, assigning higher weight to hits that occur at higher ranks. Higher values for both metrics indicate better performance. We report the average scores and their standard derivations on the testing set over three random runs. To assess the effectiveness of the proposed model with varying dataset sizes, we evaluate its performance on different proportions of the dataset: 30%, 50%, 70%, 90%, and 100%. The extraction of different dataset proportions is based on sampling patient-phenotype edges in the graph $\mathcal{G}_p$. Specifically, for each patient, we sample the required proportion of edges connected to phenotypes.

---

**Algorithm 1** The Training Procedure of MPI

---

1: **Input**: Patient multi-modal data $\mathbf{X}^M$, Codebook $\mathcal{C}$, Encoder $\mathbf{E}$, Decoder $\mathbf{D}$, GCNs;
2: **Output**: Patient representation $\mathbf{H}$;
3: **for** each iteration **do**
4:     **for** each patient $\mathbf{x}^m$ in $\mathbf{X}^M$ **do**
4:         $\mathbf{z}^m := \mathbf{E}(\mathbf{x}^m)$;
4:         Retrieve disentangled biological factors $(c_0, \ldots, c_{l-1})$ from Codebook $\mathcal{C}$;
4:         Obtain the quantized vector $\hat{\mathbf{z}}^m := \sum_{d=0}^{l-1} \mathbf{e}_{c_d}$;
4:         Reconstruct the input $\mathbf{x}^m$ based on $\hat{\mathbf{x}}^m = \mathbf{D}(\hat{\mathbf{z}}^m)$;
5:     **end for**
5:     Optimize loss in Eq.(1);
6: **end for**
6: Obtain the disentangled biological factors $\mathcal{C}$;
6: Patient-Phenotype Graph $\mathcal{G}_p$ Construction;
6: Patient-Factor Graph $\mathcal{G}_f$ Construction;
7: **for** each iteration **do**
7:     Learn node representation $\mathbf{H}_p$ and $\mathbf{H}_f$ for graph $\mathcal{G}_p$ and $\mathcal{G}_f$ using GCNs in Eq.(2), respectively;
7:     Optimize loss in Eq.(3) and Eq.(4);
8: **end for**

---

## A.4 Time Complexity

The time complexity of data quantization for each patient is composed of three primary components. The first component is the encoder, which has a time complexity of $O(D \cdot F)$, where $D$ represents the input dimensionality and $F$ is a factor that depends on the number of layers and the operations performed within the encoder. The second component is vector quantization, with a time complexity of $O(K \cdot d)$, where $K$ denotes the number of entries in the codebooks and $d$ represents the dimensionality of latent embeddings. The third component is the decoder, which has a time complexity of $O(d \cdot G)$, where $G$ is a factor related to the number of layers and operations in the decoder. Consequently, the overall time complexity of data quantization can be expressed as $O(D \cdot F + K \cdot d + d \cdot G)$. Given that both the encoder and the decoder in this study are implemented as multi-layer perceptrons (MLPs), we simplify the expression to $O(D \cdot d + K \cdot d + d^2)$ for ease of calculation. Then the updating of GNNs in each iteration mainly involves the updating of node vectors and weight matrices, whose time complexity is $O(n_t \cdot d^2 + z \cdot d)$, where $n_t = n + m$ and $z$ are the total number of nodes and the total number of edges in graph $\mathcal{G}_f$ and $\mathcal{G}_p$, respectively. $d$ is the embedding dimensionality. Lastly, the time complexity of cross-view contrastive knowledge distillation for each patient is $O(d \cdot N)$ where $N$ denotes the number of negative patients. Therefore the time complexity of MPI is $O(D \cdot d + K \cdot d + d^2 + n_t \cdot d^2 + z \cdot d + d \cdot N)$. Since $K \ll D$, $d \ll D$, $N \ll D$, and $D \ll z$, the time complexity simplifies to $O(n_t \cdot d^2 + z \cdot d)$ which is linear with $(n_t \cdot d^2 + z \cdot d)$, depending on the number of nodes and edges in the constructed graphs. It is well-known that canonical GCNs are not characterized by high time complexity, indicating the efficiency and scalability of our model.

## A.5 Limitations

The current research primarily focuses on two modalities. Future work will explore the incorporation of additional modalities. Another limitation is the selected patient cohort, as this study concentrates on Alzheimer's disease and related dementias. To enhance the generalizability of our findings, we aim to apply the proposed model to a broader range of patient cohorts and various downstream tasks.

## A.6 Broader Impacts

Phenotype imputation using biological data can advance healthcare by enabling a deeper understanding of diseases and patients' health states. It helps aids early diagnosis and personalized treatments, leading to better health outcomes. However, there are potential risks, including the possibility of exacerbating health disparities if data is not diverse, and the risk of inaccurate imputations leading to erroneous conclusions.

