# OpenReview forum: "Unified Insights: Harnessing Multi-modal Data for Phenotype Imputation via View Decoupling"
_NeurIPS.cc/2024/Conference — NeurIPS 2024 poster_

### Official Review · Reviewer_dJb6 · 2024-07-10

**Soundness:** 4
**Presentation:** 3
**Contribution:** 4
**Rating:** 7
**Confidence:** 5

**Summary:**

This paper focuses on the task of phenotype imputation and proposes utilizing multi-modal data to gain insights that facilitate the evaluation of patients' overall health status. Specifically, the authors design a framework based on view decoupling, which involves segregating the modeling of biological data and phenotype data to avoid the impact of data heterogeneity and view conflict. To alleviate the influence of noise and irrelevant information in the biological data, a novel contrastive knowledge distillation method is proposed. Furthermore, the authors conduct extensive experiments to demonstrate the superiority of the proposed model.

**Strengths:**

1.	The paper addresses a crucial problem in healthcare and is novel in combining omics data with clinical data in healthcare studies.
2.	It is useful and practical considering the prevalence and growth of biobanks, and the authors conducted experiments on a well-known biobank dataset.
3.	The applied model is easy to use and extendable to different data modalities.
4.	The proposed model is novel, and the proposed data quantization provides a feasible way to effectively model the relationship between patients from the omics data. Cross-view knowledge distillation involving intra-view and inter-view distillation improves the utilization of omics data and avoids noisy and irrelevant information.
5.	The writing of the paper is good, and it provides a comprehensive related work that helps readers better understand the task.
6.	The experiments show that the proposed model can effectively leverage omics data and demonstrate improved imputation performance on a real biomedical database. Extensive ablation studies and analyses provide more insights and help understand the model design.

**Weaknesses:**

1.	The model includes multiple components. It would be beneficial to discuss the time complexity of the proposed method. Specifically, an analysis of the computational efficiency for each component, as well as the overall model, would provide valuable insights.
2.	The patients in the experiments are selected from those with Alzheimer's disease and related dementias. It would be helpful to explain the rationale behind selecting this particular patient set. Additionally, it is important to discuss whether the model is applicable to other cohorts.
3.	Why can't recent models, such as M3Care, Graph, and MUSE, directly address the need for integrating biological data and EHR data?
4.	The proposed method involves multiple loss functions. Adding these losses to Figure 1 would aid understanding. Including pseudocode for the algorithm would also be helpful.

**Questions:**

Please refer to weaknesses.

**Limitations:**

Please refer to weaknesses.

---

> ### Author Rebuttal · Authors · 2024-08-07
>
> Dear Reviewer dJb6
>
> Thank you for the review and valuable comments. We respond to your questions below.
>
> **1. Time complexity.**
>
> The time complexity of data quantization for each patient is composed of three primary components. The first component is the encoder, which has a time complexity of $O(D \cdot F)$, where $D$ represents the input dimensionality and $F$ is a factor that depends on the number of layers and the operations performed within the encoder. The second component is vector quantization, with a time complexity of $O(K \cdot d)$, where $K$ denotes the number of entries in the codebooks and $d$ represents the dimensionality of latent embeddings. The third component is the decoder, which has a time complexity of $O(d \cdot G)$, where $G$ is a factor related to the number of layers and operations in the decoder. Consequently, the overall time complexity of data quantization can be expressed as $O(D \cdot F + K \cdot d + d \cdot G)$. Given that both the encoder and the decoder in this study are implemented as MLPs, we simplify the expression to $O(D \cdot d + K \cdot d + d^2)$ for ease of calculation.
>
> Then the updating of GNNs in each iteration mainly involves the updating of node vectors and weight matrices, whose time complexity is $O(n_t\cdot d^2 + z\cdot d)$, where $n_t $ and $z$ are the total number of nodes and the total number of edges in graph $\mathcal{G}_f$ and $\mathcal{G}_p$, respectively. $d$ is the embedding dimensionality.
>
> Lastly, the time complexity of cross-view contrastive knowledge distillation for each patient is $O(d \cdot N)$ where $N$ denotes the number of negative patients.
>
> Therefore the time complexity of MPI in each iteration is $O(D \cdot d + K \cdot d + d^2 + n_t\cdot d^2 + z\cdot d + d \cdot N)$. Since $K \ll D$, $d \ll D$, $N \ll D$, and $D \ll z$, the time complexity simplifies to $O(n_t\cdot d^2 + z\cdot d)$ which is linear with $(n_t\cdot d^2 + z\cdot d)$, depending on the number of nodes and edges in the constructed graphs. It is well-known that canonical GCNs are not characterized by high time complexity, indicating the efficiency and scalability of our model.
>
> **2. Why Alzheimer’s disease and related dementias**
>
> Chronic diseases, especially neurodegenerative diseases, frequently exhibit missing phenotypes due to mild or nonspecific initial symptoms. Routine data collection processes might overlook these subtle signs until more pronounced symptoms emerge. This can be particularly challenging in the context of neurodegenerative diseases like Alzheimer’s Disease and Related Dementias (ADRD), where early detection is crucial for timely intervention and management. Furthermore, research on ADRD particularly emphasizes early detection and intervention, which aligns well with the research goals of identifying historical phenotypes.
>
> Additionally, the choice of an ADRD cohort involves a relatively larger cohort compared to other neurodegenerative diseases. Alzheimer’s is one of the most common neurodegenerative diseases, which ensures that the data includes a wide range of phenotypic expressions and stages of disease. This diversity is crucial for studying the full spectrum of phenotype presentation and identifying underlying missing signs.
>
> **3. M3Care, Grape, and MUSE**
>
> These models can be applied to integrate biological data and EHR data. However, they are often inferior to the naive combination of clinical and biological data. This is due to the conflict between clinical and biological views. In detail, GRAPE uses each feature dimension as a node, which is not suitable for high-dimensional feature imputation. M3Care computes patient similarity for each modality separately, thereby failing to explore cross-modality correlations. MUSE connects patients to only two modality nodes, introducing noisy edges between patients. These methods are not specifically designed to integrate biological data with EHR data, highlighting the importance and unique capability of our model.
>
> **4. Figure 1 and pseudocode**
>
> Thanks for your suggestion. We will revise Figure 1 and add pseudocode in the Appendix.
>
> We hope our explanation can solve your concerns. If you have any other questions or concerns, please feel free to let us know and we are more than happy to answer and make clarifications.

---

> > ### Comment · Reviewer_dJb6 · 2024-08-13
> >
> > Thank you for the detailed response. The authors have addressed all my concerns. I would like to maintain my positive rating.

---

### Official Review · Reviewer_W1JA · 2024-07-10

**Soundness:** 4
**Presentation:** 3
**Contribution:** 3
**Rating:** 7
**Confidence:** 5

**Summary:**

The work integrates multi-modal biological data into the task of phenotype imputation, addressing the challenges of heterogeneity and imprecision inherent in fusing biological and phenotype data. This paper introduces the MPI model, a novel approach that incorporates multi-modal data through a view decoupling strategy. By modeling patients from both biological and phenotype perspectives, the model preserves knowledge from each view. Subsequently, a cross-view contrastive knowledge distillation technique is proposed to enhance phenotype imputation. Extensive experiments demonstrate the superiority of MPI compared to state-of-the-art methods.

**Strengths:**

Originality:
-	Incorporating multi-modal biological data for phenotype imputation is well-motivated and important. As far as I know, this is the first paper to leverage biological data to improve phenotype imputation. Biological information usually cannot be directly leveraged in clinical analysis in current studies, the model finds a novel way to integrate biological data with EHRs by modeling them using codebooks and graph representation learning.
-	In addition, the authors also identify and address the essential heterogeneity and imprecise issues in biological data, which make the model applicable in real-world clinical studies.
-	The proposed method is tailored to the motivation and tries to learn from separate views and preserve comprehensive knowledge to represent patients. The novelty of the methodology is significant; the proposed data quantization method could learn the latent common factors of patients and further model the correlation from the biological view.
Quality
-	The proposed method is sound and designed to solve the heterogeneity and imprecision issues.
-	The overall model structure is well-designed, not complex, and easy to apply in real-world tasks.
Clarity
-	The paper is well-written and easy to follow.
-	The paper is also well-organized.
Significance
-	This paper conducts extensive experiments and comprehensive ablation studies to evaluate the proposed model. The comparison with the most recent baselines shows the effectiveness of the proposed method, especially with varying dataset proportions.

**Weaknesses:**

-	The paper focuses on the utilization of multi-modal biological data. In experiments, the authors leverage two modalities: Proteomics and Metabolomics. It is expected that the model can be applied to more modalities.

-	The proposed model needs to learn codebooks to model the correlation of patients. The quality of the learned codebook could influence the imputation performance since imputation is based on the representation learned from the graph where latent factors serve as nodes.

**Questions:**

1.	Can the proposed model be applied to other modalities if available?
2.	Can the authors provide more discussion on the codebook setting and its impact

**Limitations:**

The limitation has been discussed in Appendix.

---

> ### Author Rebuttal · Authors · 2024-08-07
>
> Dear Reviewer W1JA,
>
> Thank you for the review and valuable comments. We respond to your questions below.
>
> **1. Other modalities.**
>
> Yes, our method can be certainly applied to other modalities if data is available. For each modality, a distinct encoder and decoder can be utilized to uncover the latent factors specific to that modality. Subsequently, a patient-factor graph can be constructed alongside the patient-phenotype graph. Predictions can then be made based on the representations learned by GNN models.
>
> **2. Codebook setting and its impact**
>
> The quality of the learned codebooks indeed has an impact on the model performance. For example, a smaller number of codebooks, such as one or two, may fail to capture sufficient fine-grained information from the biological data. Conversely, larger codebooks might introduce additional underlying factors due to finer granularity, which could reduce their discriminative power for patient profiling. In addition, smaller codebook sizes may fail to capture some underlying biological factors, resulting in insufficient information for patient profiling. Conversely, larger codebook sizes might lead to certain codes being underutilized, which can hinder the overall optimization of the codebook. Both the number of codebooks and the codebook size can affect the quality of the learned codebooks. The empirical results in Figure 3 demonstrate the effect of different codebook settings.
>
> We hope our explanation can solve your concerns. If you have any other questions or concerns, please feel free to let us know and we are more than happy to answer and make clarifications.

---

### Official Review · Reviewer_8HRK · 2024-07-12

**Soundness:** 3
**Presentation:** 3
**Contribution:** 2
**Rating:** 4
**Confidence:** 5

**Summary:**

This paper introduces a machine learning (ML) approach aimed at addressing the challenge of phenotype missing data in clinical datasets. Specifically, the authors propose utilizing multi-modal biological data to enhance patient health information, thereby improving the accuracy of phenotype imputation. To accommodate the heterogeneity and inherent imprecision of multi-modal biological data, they introduce the Multimodal data for Phenotype Imputation (MPI) framework. Initially, the authors employ a data quantization technique to denoise biological data, focusing on learning latent biological factors. Subsequently, they suggest modeling biological and phenotype data separately, followed by aligning their information using contrastive learning methods. The authors validate the efficacy of their approach for phenotype imputation using real-world clinical data.

**Strengths:**

- This paper addresses the issue of missing data in clinical contexts, a serious obstacle that impedes the predictive performance of AI models in healthcare.
- Utilizing biological data to enhance patient health information is logical.

**Weaknesses:**

- The model design lacks clarity and motivation. Specifically, it is unclear why residual quantization is considered suitable for reducing noise in biological data. The authors should provide a more detailed justification for this design choice and conduct additional experiments to verify the effectiveness of residual quantization compared to other denoising methods. For example, auto-encoder without quantization?

- The technical contribution of the proposed method appears limited. For instance, Graph Convolutional Networks (GCN) are a standard method for capturing information from graph-structured data, and contrastive learning is a common approach for aligning representations from multi-modal data. Moreover, is there a specific reason for choosing GCN over more advanced and recent graph neural network methods such as Graph Isomorphism Network (GIN) [1]?

- The UK Biobank is a comprehensive clinical dataset that includes diverse biological data such as genomics, biochemical markers, haematological markers, infectious disease markers, metabolomics, telomere measures, etc. Is there a rationale for focusing only on proteomics and metabolomics in this study? Furthermore, why were Alzheimer’s disease and related dementias chosen as the focus of this study?

- The authors claim that their proposed ranking loss outperforms classification loss for phenotype imputation. An ablation study should be conducted to substantiate this claim.

- In addition to benchmarking phenotype imputation, the authors should perform additional experiments to demonstrate the benefits of their proposed phenotype imputation method in downstream clinical tasks compared to baseline methods.

References

[1] Xu, Keyulu, et al. "How powerful are graph neural networks?." ICLR 2019.

**Questions:**

See the above section.

**Limitations:**

N/A. The limitation section is just about the study design, which is not the potential negative societal impact of their work.

---

> ### Author Rebuttal · Authors · 2024-08-07
>
> Dear Reviewer 8HRK,
>
> Thank you for the review and valuable comments. We respond to your questions below.
>
> **1. Residual quantization**
>
> We respectfully highlight that our proposed biological data quantization is not for reducing noise in biological data.
>
> The motivation behind residual quantization as we discussed in the introduction is that the biological conditions of patients uncover major underlying factors that indicate health status. In other words, patients sharing similar underlying biological factors could have similar phenotypes. Identifying these latent factors would facilitate the effective characterization of patients and their phenotypes. Therefore, we propose quantizing the biological data and uncovering the corresponding factors using Residual Quantization. Then we are able to model the correlation between patients from a biological view and learn patient representations by modeling the relationship between patients and latent factors in a patient-factor graph.
>
> Comparison with AE: Applying an Autoencoder (AE) to encode the biological data results in extracting a single low-dimensional biological feature for each patient. This feature serves as the initial embedding of patients in a GCN. By using GCN on the patient-phenotype graph, we can then obtain patient representations that enable the prediction of missing phenotypes based on these learned representations. The result comparison can be found in the table below. We can observe that roughly using the encoded biological data as the feature cannot achieve better performance compared with MPI.
>
> | % | Metric | AE+GCN | MPI |
> | --- | --- | --- | --- |
> | 50% | H@10 | 27.25 | 31.28 |
> |  | H@20 | 42.73 | 47.55 |
> |  | H@50 | 69.04 | 72.99 |
> |  | MRR | 12.28 | 14.83 |
> | 70% | H@10 | 31.06 | 35.68 |
> |  | H@20 | 46.79 | 51.59 |
> |  | H@50 | 71.28 | 75.82 |
> |  | MRR | 14.16 | 17.44 |
> | 100% | H@10 | 35.32 | 38.74 |
> |  | H@20 | 51.65 | 55.10 |
> |  | H@50 | 74.68 | 78.42 |
> |  | MRR | 16.31 | 19.28 |
>
> **2. The technical contribution**
>
> We would like to highlight that our main technical contributions lay in:
>
> 1) The view decoupling strategy, which separates the modeling of multi-modal biological data from phenotype data to address the conflicts arising from their joint modeling.
>
> 2) The latent biological factor uncovering in patients through data quantization, which Identifies these latent factors enables more effective characterization of patients and their phenotypes.
>
> 3) The cross-view contrastive knowledge distillation, which builds upon traditional contrastive learning by introducing a novel approach that incorporates cross-view information with intra-view and inter-view negative pairs.
>
> Additionally, we want to clarify that the graph neural model is not the focus of our paper. We use GCNs as a basic backbone model, which can be substituted with any existing advanced graph neural network. Our emphasis is on demonstrating that the improvements in our model primarily stem from our designed components rather than the sophistication of the GNN itself. Therefore, we opted to use the basic GCN model.
>
> **3. Proteomics and metabolomics**
>
> We leveraged proteomics and metabolomics as biological studies have pointed out the associations and predictive value of plasma proteomics and metabolomics with ADRD. Since biological processes could begin years before the onset of clinical symptoms, proteomics and metabolomics which comprise the end-product of genes, transcripts, and protein regulations, offer insight into identifying alterations in multiple biochemical processes and the risk of ADRD among cognitively healthy adults [1,2]. The model could be certainly applied to other modalities with a corresponding pre-trained encoder.
>
> [1] Plasma metabolomic profiles of dementia: a prospective study of 110,655 participants in the UK Biobank
>
> [2] Proteome Network Analysis Identifies Potential Biomarkers for Brain Aging. Journal of Alzheimer's Disease, 2023.
>
> **4. Why Alzheimer’s disease and related dementias**
>
> Chronic diseases, especially neurodegenerative diseases, frequently exhibit missing phenotypes due to mild or nonspecific initial symptoms. Routine data collection processes might overlook these subtle signs until more pronounced symptoms emerge. This can be particularly challenging in the context of neurodegenerative diseases like Alzheimer’s Disease and Related Dementias (ADRD), where early detection is crucial for timely intervention and management. Furthermore, research on ADRD particularly emphasizes early detection and intervention, which aligns well with the research goals of identifying historical phenotypes.
>
> Additionally, the choice of an ADRD cohort involves a relatively larger cohort compared to other neurodegenerative diseases. Alzheimer’s is one of the most common neurodegenerative diseases, which ensures that the data includes a wide range of phenotypic expressions and stages of disease. This diversity is crucial for studying the full spectrum of phenotype presentation and identifying underlying missing signs.
>
> **5. Downstream tasks**
>
> Thanks for your suggestion. In this paper, we concentrate on the algorithmic development for phenotype imputation. In future work, we will adapt and validate our model for various downstream clinical tasks.
>
> **6. Ranking loss**
>
> We included classification loss as a variant in our implementation, yet experimental results show that ranking loss outperforms classification loss (see below). This is likely because ranking loss compares the similarity between positive and negative edge pairs rather than showing class probabilities. Given the individual specificity in phenotype imputation, ranking loss better captures the hidden patterns in patients' phenotypes.
>
> |100\% | Hits@10 | Hits@20 | Hits@50 | MRR   |
> | --- | --- | --- | --- | --- |
> | MPI w/ classification | 36.49 |  53.76  | 77.28 | 18.32    |
> | MPI w/ ranking | 38.74 |  55.10  | 78.41 | 19.27    |

---

### Official Review · Reviewer_QunD · 2024-07-13

**Soundness:** 3
**Presentation:** 3
**Contribution:** 2
**Rating:** 5
**Confidence:** 4

**Summary:**

This paper introduces MPI, a framework designed to improve phenotype imputation in EHR by leveraging multi-modal biological data through view decoupling. The method consists of quantizing biological data to identify the latent factors, using them to create a correlation graph, and a separate graph which models phenotypic co-occurrence. Moreover, a cross-view contrastive knowledge distillation strategy is employed to better leverage the noisy and sparse biological data.

**Strengths:**

The MPI framework handles the sparsity in the multi-modal biological data and effectively utilizes them to provide a more comprehensive understanding of patient profiles for better phenotype imputation.

**Weaknesses:**

The complexity of the proposed MPI method makes it challenging to interpret the results and understand the underlying biological signals driving the phenotype imputation. This could be important for clinical applications.

**Questions:**

1. How does the sparsity/missing rates of the biological data affect the performance of MPI?
2. How is each modality data (proteomics and metabolomics) pre-processed before feeding into the encoders? This information is not clearly described in the appendix.

**Limitations:**

The performance of MPI might highly depend on the size of the dataset.

---

> ### Author Rebuttal · Authors · 2024-08-07
>
> Dear Reviewer QunD,
>
> Thank you for the review and valuable comments. We respond to your questions below.
>
> **1. Sparsity/missing rates**
>
> The biological modalities used in this study exhibit significant missingness at random, with approximately 90% missingness in proteomics and 50% in metabolomics. The proposed MPI demonstrates superior performance compared to baseline methods in handling extreme missingness. Given the high missing rates, it is impractical to conduct experiments with varied missing rates. However, we evaluate MPI through ablation studies that utilize each modality independently, representing scenarios with complete missingness in the respective modalities.
>
> **2. Modality data pre-processing**
>
> Proteomics data, including levels of around 3,000 proteins, are provided as Normalized Protein eXpression (NPX) values, obtained after UK Biobank preprocessing, which includes median centering normalization between plates and log2 transformation. We used these NPX values directly as the encoder input without further processing [1]. For metabolomics, we applied a natural logarithmic transformation (ln[x+1]) to all metabolite values, followed by Z-transformation [2].
>
> [1] Chen, Lingyan, et al. "Systematic Mendelian randomization using the human plasma proteome to discover potential therapeutic targets for stroke." Nature Communications 13.1 (2022): 6143.
>
> [2] Zhang, Xinyu, et al. "Plasma metabolomic profiles of dementia: a prospective study of 110,655 participants in the UK Biobank." BMC Medicine 20.1 (2022): 252.
>
> We hope our explanation can solve your concerns. If you have any other questions or concerns, please feel free to let us know and we are more than happy to answer and make clarifications.

---

### Official Review · Reviewer_wLvH · 2024-07-21

**Soundness:** 2
**Presentation:** 3
**Contribution:** 2
**Rating:** 4
**Confidence:** 3

**Summary:**

This paper proposes the MPI framework for phenotype imputation. Focusing on the detrimental effects of heterogeneity and inaccuracies in phenotype imputation, the proposed framework separates the biological view and the phenotype view for model learning and integrates them afterward. In experiment results using clinical datasets and ablation studies, it is shown that the proposed approach outperforms existing methods in the accuracy of phenotype imputation and highlight the advantages of separating the biological view and the phenotype view in model learning.

**Strengths:**

This paper proposes a framework that separates and integrates from the biological view and the phenotype view for phenotype imputation.
It is important to estimate missing phenotypes from the perspective of promoting more appropriate medical practice. The well-thought-out structure of sections and subsections makes the proposed method easy to understand.

**Weaknesses:**

- The evaluation of the effectiveness of the proposed method is weak because a general graph neural model was not included as a comparative method.
- The claim that "By applying this cross-view contrastive optimization, our model effectively captures the intricate relationships within both the biological and collaborative views, leading to robust representations of the patients" is not supported by experiments or other evidence.
- There is no discussion on the validity of the predicted phenotypes.
- There are doubts about the content of the experiments and the ablation study.

**Questions:**

- Why were the general graph neural models [33, 37, 46], which were referenced for the experimental setup, not included as comparative methods?
- Are there any references supporting the claim that "The existing approaches fail to recognize and disentangle the heterogeneous factors present in biological data"?
- How did you determine the value of the margin hyperparameter γ in experiments?
- Why did you only conduct experiments on Alzheimer’s disease and related dementia datasets? For example, diabetes is also one of the chronic diseases.
- Patients with "the rare phenotypes" and "the most prevalent phenotypes" were excluded from dataset. I believe these should not be excluded for the purpose of phenotype imputation. Is there a more compelling reason for their exclusion?
- Why is the variance not included in the figures of the ablation study results?
- The structure of the V4 model is unclear. What kind of model is it?
- Is it possible to show a table like Table 2 for the results of the ablation study in appendices? I think that the results of ablation study is shown a portion of the entire experiment.

**Limitations:**

Yes. In their appendix, the authors describe the limitations and potential negative societal impact of their work.

---

> ### Author Rebuttal · Authors · 2024-08-07
>
> Dear Reviewer wLvH,
>
> Thank you for the review and valuable comments. We respond to your questions below.
>
> **1. General graph neural models**
>
> We respectfully highlight that our experiments compared general graph neural models, specifically GraphSage and GIN. The models you referred to are designed for recommendation systems and path-based link prediction, and thus, are not general graph neural models. They are used as references for dataset partitioning to demonstrate the rationale behind the dataset division.
>
> **2. Cross-view contrastive optimization**
>
> We respectfully highlight that we provided experimental evidence in the ablation study to support this claim. Specifically, in Table 3, V4 is a variant model without cross-view contrastive optimization, whereas MPI incorporates this design. As observed in Table 3, MPI consistently outperforms V4, demonstrating that our cross-view contrastive optimization effectively enables robust patient representation learning.
>
> **3. Validity**
>
> Thanks for your question. In this paper, we concentrate on the algorithmic development for phenotype imputation and demonstrate the model’s superiority against baselines based on the selective metrics. In future work, we will adapt and validate our model for various downstream clinical tasks.
>
> **4. Supporting References**
>
> Thanks for pointing it out. We will add references in the revision. Current methods simply treat biological data as features and employ canonical machine learning techniques to encode them, failing to disentangle the heterogeneous factors.
>
> - Imputation of label-free quantitative mass spectrometry-based proteomics data using self-supervised deep learning, Nature Communication 2024
>
> - Toward an integrated machine learning model of a proteomics experiment, Journal of Proteome Research, 2023
>
> - Recent advances in mass spectrometry-based computational metabolomics, Current Opinion in Chemical Biology, 2023
>
> **5. Determine the value of the margin hyperparameter**
>
> The margin hyperparameter 𝛾 is determined through a search in the set {1, 3, 5,10}. We will include this detail in the implementation supplements.
>
> **6. Only Alzheimer’s disease and related dementia dataset**
>
> Chronic diseases, especially neurodegenerative diseases, frequently exhibit missing phenotypes due to mild or nonspecific initial symptoms. Routine data collection processes might overlook these subtle signs until more pronounced symptoms emerge. This can be particularly challenging in the context of neurodegenerative diseases like Alzheimer’s Disease and Related Dementias (ADRD), where early detection is crucial for timely intervention and management. Furthermore, research on ADRD particularly emphasizes early detection and intervention, which aligns well with the research goals of identifying historical phenotypes.
>
> Additionally, the choice of an ADRD cohort involves a relatively larger cohort compared to other neurodegenerative diseases. Alzheimer’s is one of the most common neurodegenerative diseases, which ensures that the data includes a wide range of phenotypic expressions and stages of disease. This diversity is crucial for studying the full spectrum of phenotype presentation and identifying underlying missing signs.
>
> Diabetes typically involves more straightforward clinical measurements (e.g., blood glucose levels, HbA1c) and may not have the same challenges as ADRD. In the future, we will identify additional chronic diseases suitable for the task to evaluate our method.
>
> **7. Patients with the rare phenotypes and the most prevalent phenotypes were excluded from the dataset.**
>
> We filter out phenotypes with an occurrence of less than 20 while our cohort population reaches about 15000. The small occurrence (0.06%) reflects the less practical value in this work of imputing these phenotypes. Meanwhile, there are a few phenotypes with quite high frequency, e.g., more than 4000 individuals with hypertension. Since ADRD generally focuses on the elderly population, these phenotypes are typically with less specificity and regarded as possible confounders due to aging. Meanwhile, these phenotypes might dominate the dataset, obscuring other important associations, while focusing on moderately prevalent phenotypes could reveal more subtle associations.
>
> **8. Variance not in the ablation study results**
>
> We did not report the variance for the ablation study in Table 3 due to the table space constraints. However, we are happy to include the variance in the Appendix during the revision.
>
> **9. The structure of the V4 model**
>
> We employ the same GCN used in MPI as the V4 model. The key difference is that the V4 model operates on a single homogeneous graph that integrates biological factors, patients, and phenotypes, whereas MPI utilizes two separate graphs to model patients from both phenotype and biological perspectives. Note that both MPI and V4 models are based on biological factors derived from Biological Data Quantization.
>
> **10. Show a table like Table 2 for the results of the ablation study**
>
> We are happy to reframe Table 3 of the ablation study to match the format of Table 2 in the Appendix. Thanks for the suggestion.
>
> We hope our explanation can solve your concerns. If you have any other questions or concerns, please feel free to let us know and we are more than happy to answer and make clarifications.

---

### Author Response · Authors · 2024-08-12
**A General Response by Authors**

Dear Reviewers,

We would like to thank all the reviewers for their thoughtful suggestions on our paper.

In the rebuttal period, we have provided detailed responses to all the comments and questions point-by-point. We would appreciate all reviewers’ time again. Would you mind checking our response and confirming whether you have any further questions?

We are anticipating your post-rebuttal feedback!

Best

Authors

---

### Decision · Program_Chairs · 2024-09-25

**Decision:**

Accept (poster)

**Comment:**

The authors propose MPI for phenotype imputation using multimodal data. To decrease noise, they quantize and embed the biological signals. Then they constract two graphs and use graph neural networks to obtain representations. Finally, they use Cross-view Contrastive Knowledge Distillation by regarding the biological-view graph encoder as the teacher model and the phenotype-view graph encoder as the student model.

The reviewers have praised the practicality of the proposed method. However, there are concerns about the complexity of the algorithm, which is, to a degree, unavoidable when dealing with multimodal data. The reviewers have also provided feedback about the evaluation and I recommend the authors to address them in the camera-ready version.